# Pooled prevalence and associated factors of pregnancy termination among youth aged 15–24 year women in East Africa: Multilevel level analysis

**Samuel Hailegebreal** [1] *, **Ermias Bekele Enyew** [2], **Atsedu Endale Simegn** [3], **Binyam Tariku Seboka** [4], **Girma Gilano** [1], **Reta Kassa** [5], **Mohammedjud Hassen Ahmed** [2], **Yosef Haile** [6], **Firehiwot Haile** [6]

1 Department of Health Informatics, College of Medicine and Health Science, School of Public Health, Arba Minch University, Arba Minch, Ethiopia, 2 Department of Health Informatics, Institute of Public Health, Mettu University, Metu, Ethiopia, 3 Department of Anesthesia, College of Medicine and Health Science, Wachemo University, Hosaena, Ethiopia, 4 Department of Health Informatics, College of Medicine and Health Sciences, Dilla University, Dilla, Ethiopia, 5 School of public health, Dilla University, Dilla, Ethiopia, 6 Arba Minch University, College of Medicine and Health Sciences, School of Public Health, Arba Minch, Ethiopia

* samuastd@gmail.com

## Abstract

### Background

Most of unwanted pregnancies among adolescent girls and young women (AGYW) in Africa result in pregnancy termination. Despite attempts to enhance maternal health care service utilization, unsafe abortion remains the leading cause of maternal death in Sub-Saharan Africa (SSA), there is still a study gap, notably in East Africa, where community-level issues are not studied. Therefore, this study aimed to assess pooled prevalence pregnancy termination and associated factors among youth (15–24 year-old) women in the East Africa.

### Methods

The study was conducted based on the most recent Demographic and Health Surveys (DHS) in the 12 East African countries. A total weighted sample of 44,846 youth (15–24) age group women was included in this study. To detect the existence of a substantial clustering effect, the Intra-class Correlation Coefficient (ICC), Median Odds Ratio (MOR), and Likelihood Ratio (LR)-test were used. Furthermore, because the models were nested, deviance (-2LLR) was used for model comparison. In the multilevel logistic model, significant factors related to pregnancy termination were declared using Adjusted Odds Ratios (AOR) with a 95%Confidence Interval (CI) and p-value of 0.05.

### Result

The pooled prevalence of pregnancy termination in East African countries was 7.79% (95% CI: 7.54, 8.04) with the highest prevalence in Uganda 12.51% (95% CI: 11.56, 13.41) and lowest was observed in Zambia 5.64% ((95% CI: 4.86, 6.41). In multilevel multivariable

dhsprogram.com/data/dataset_admin/login_main.
cfm after reasonable request of the program.

**Funding:** The authors received no specific funding
for this work.

**Competing interests:** The authors declare that they
have no competing interests.

**Abbreviations:** AOR, Adjusted Odds Ratios; CI,
Confidence Interval; CSA, Central Statistical
Agency; DHS, Demographic and Health Survey;
ICC, Intra-class Correlation Coefficient; LLR, Log-
Likelihood Ratio; MOR, Median Odds Ratio.

logistic regression result, age 20–24 [AOR = 1.93; 95% CI: 1.71, 2.16], media exposure [AOR = 1.22; 95% CI: 1.12, 1.34], married [AOR = 1.32, 95% CI: 1.21, 1.43], had working [AOR = 1.13; 95% CI: 1.04, 1.23],no education[AOR = 3.98, 95% CI: 2.32, 6.81], primary education [AOR = 4.05, 95% CI: 2.38, 6.88], secondary education [AOR = 2.96, 95% CI: 1.74, 5.03], multiparous [AOR = 0.85; 95%CI: 0.79, 0.93], sexual initiation greater or equal to 15 [AOR = 0.82; 95%CI: 0.74, 0.99] were significantly associated with pregnancy termination.

## Conclusion

The pooled prevalence of pregnancy termination in East Africa was high in this study. Maternal age, marital status, education status, parity, age at first sex, media exposure, working status and living countries were significantly associated with pregnancy termination. The finding provides critical information for developing health interventions to decrease unplanned pregnancies and illegal pregnancy termination.

## Background

Pregnancy termination is defined as a pregnancy that is terminated by choice through intervention [1]. Every year, over 73 million induced abortions are performed around the world [2]. Six out of ten unwanted pregnancies (61%) and three out of ten (29%) of all pregnancies result in induced abortion [2]. According to global estimates from 2010 to 2014, 45% of all induced abortions are unsafe in the world. In Africa, one-third of all unsafe abortions were carried out under the most risky conditions, i.e. by untrained people using harmful and invasive procedures [3].

Developing countries account for 97% of all unsafe abortions. More than half of all unsafe abortions happen in Asia, with the majority taking place in South and Central Asia. Nearly half of all abortions in Africa arise in the most dangerous conditions [4]. In Sub-Saharan Africa (SSA), 57% of young women between the ages of 15–24 years have pregnancy termination [5]. The proportion of pregnancies ending in abortion varies from 12% in Western Africa to 24% in Southern Africa; rates in Middle, Eastern and Northern Africa are 13%, 14% and 23%, respectively [1].

Unsafe abortion is a primary cause of maternal death and morbidity, but it can be avoided. It can cause physical and mental health problems, as well as social and financial hardships for women, communities, and health-care systems [3,6,7]. Globally, each year, 4.7%–13.2% of maternal deaths can be attributed to unsafe abortion [8]. In developing regions, that number rises to 220 deaths per 100 000 unsafe abortions [4]. According to estimates from 2012, 7 million women were treated in hospitals in developing countries for problems related to unsafe abortions each year [9]. Africa has the greatest rate of abortion-related mortality in the world. In Africa, unsafe abortion caused at least 9% of maternal deaths (or 16,000 deaths) in 2014 [10]. Previous studies revealed that maternal age [11,12], parity [13,14], occupation [15], age at first sex [13,16], marital status[17,18], place of residence [15], Media exposure [19], wealth index [15,20], education status [21,22] and region [23,24] were significantly associated factor with pregnancy termination.

As far as our literature search nothing is known about the pooled prevalence of pregnancy termination and associated factors among youth 15–24 year-old women in the East Africa.

Therefore, this study aimed at investigating the pooled prevalence and associated factors of pregnancy termination in East African Countries based on the most recent Demographic and Health Surveys (DHSs). The study's findings would be useful for evidence-based country-specific interventions to improve maternal health and re-align policy directions toward achieving one of the UN Sustainable Development Goals (i.e., reducing global maternal mortality ratio to less than 70 per 100,000 live births by 2030), and this research will try to fill the above evidence gap. salvaged.

## Methods and materials

### Data source and study setting

The Demographic and Health Survey (DHS) data were pooled from the 12 East Africa Countries from 2008 to 2017. The study was conducted based on the most recent Demographic and Health Surveys (DHS) in the 12 East African countries (Burundi, Ethiopia, Comoros, Uganda, Rwanda, Tanzania, Mozambique, Madagascar, Zimbabwe, Kenya, Zambia, and Malawi) conducted from 2008 to 2018. These datasets were combined to determine the pooled prevalence and factors associated with pregnancy termination among reproductive age group women in East Africa.

### Sampling procedures and study population

Using the Population and Housing Census (PHC) as a sampling frame, the DHS used a two-stage stratified sampling technique to select study participants. In the first stage, Enumeration Areas (EAs) were selected with probability sampling proportional to the size of the EAs with independent selection in each sampling stratum. In the second stage, households were systematically selected. The key demographic and health indicators were collected in each DHS [25]. The source population was all pregnant young women 15–24 years old across the east African countries whereas, all pregnant young women 15–24 years old in the selected enumeration areas were the study population. The data used in the analyses were weighted to explain variation in the probability of selection as well as non-response. A weighted total of **44,846** youth 15–24 age group women were included in this study, with a complete answer to all factors of interest. The detailed sampling procedure was presented in each country's DHS report.

### Variables of the study

**Outcome variable.** The outcome variable employed for this study was "pregnancy termination" which was derived from the question "have you ever had a terminated pregnancy?", and the response was coded as 0 = "No" and 1 = "Yes".

**Explanatory variables.** The explanatory variables (both individual and community) were selected on the basis of an association with the outcome reported in studies identified in a scoping review of the literature to inform the analysis, and on the availability of variables in the DHS datasets.

**The Individual-level variables** were age, educational status, occupation, parity, marital status, wealth index, age at first sex, and media exposure. The community-level factors were community-wealth index, community-level education, community-level mass media (radio, TV and newspaper) exposure).

### Operational definition

**Community-level poverty**: Proportion of women who were from households belonging to the categories of poorest and poorer wealth index. Those who fell at the median value and above

were categorized under the high poverty level and those who fell below the median value of the variables were categorized under the low poverty level.

**Community-level literacy:** the proportion of mothers who have completed primary school and above were categorized under literate, as opposed to mothers who have not completed primary school are categorized under illiterate.

**Community-level media exposure:** The proportion of women in the cluster who had at least some exposure to television, radio, or newspaper was categorized as media exposure, whereas mothers who did not have at least some exposure to television, radio, or newspaper were categorized as not media exposure.

### Data management and analysis

The statistical software STATA version 14 was used to handle and analyze the data. Before any statistical analysis, the data were weighted to restore the data's representativeness and provide a reliable estimate and standard error. Frequencies and percentages were used to create descriptive statistics. The pooled prevalence of pregnancy termination with a 95% Confidence Interval (CI) was reported and presented in a forest plot for East Africa Countries.

The DHS data had a hierarchical nature that could violate the independence of observations and the equal variance assumption of the traditional logistic regression model. This implies that there is a need to take into account the between cluster variability by using advanced models. Therefore, a multilevel logistic regression model (both fixed and random effects) was fitted. Since the outcome variable was binary, standard logistic regression and multilevel logistic regression models were fitted.

In the multilevel logistic regression model, we ran four models to estimate both the fixed effects of the individual and community-level factors and the random intercept of between-cluster variation. Since the models were nested, the Intra-class Correlation Coefficient (ICC), Likelihood Ratio (LR) test, Median Odds Ratio (MOR), and deviance (-2LLR) values were used to assess model comparison and fitness [26]. Variance inflation factor (VIF) was used to check for multi-collinearity and there was no evidence of multicollinearity. Accordingly, a mixed effects logistic regression model (fixed and random effects) was the best-fitted model since it had the lowest deviance value. In the multilevel logistic model, significant factors related to pregnancy termination were declared using Adjusted Odds Ratios (AOR) with a 95 percent Confidence Interval (CI) and p-value of 0.05.

### Ethics approval and consent to participate

Since the study was a secondary data analysis of publicly available survey data from the MEASURE DHS program, ethical approval and participant consent were not necessary for this particular study. We requested DHS Program and permission was granted to download and use the data for this study from (https://dhsprogram.com/data/dataset_admin/login_main.cfm). We confirm that all methods were carried out in accordance with the relevant guidelines and regulations.

## Result

### Socio-demographic characteristics of the respondent

A total of 44,837(weighted) of youth women were enrolled in this study. The majority of the respondents underlie in the age group of 20–24 (80.49%). About 57.67% of respondents had attained primary education and 48.08% of respondents were from poor households. The

**Table 1. Individual characteristics of the respondent in East African countries (n = 44,837).**

| Variables | Weighted frequency | Percent |
| --- | --- | --- |
| **Age (in years)** | | |
| 15–19 | 8,748 | 19.51% |
| 20–24 | 36,089 | 80.49% |
| **Educational status** | | |
| No education | 7,434 | 16.58% |
| Primary education | 25,860 | 57.67% |
| Secondary education | 11,018 | 24.57% |
| Higher education | 525 | 1.17% |
| **Media exposure** | | |
| No | 16,221 | 36.18% |
| Yes | 28,616 | 63.82% |
| **Wealth index** | | |
| Poor | 21,556 | 48.08% |
| Middle | 8,748 | 19.51% |
| Rich | 14,533 | 32.41% |
| **Marital status** | | |
| Single | 17,686 | 39.44% |
| Married | 27,151 | 60.56% |
| **Parity** | | |
| Prim parous | 18,216 | 40.63% |
| Multi parous | 26,621 | 59.37% |
| **Working status** | | |
| Not working | 18,420 | 41.08% |
| Working | 26,417 | 58.92% |
| **Age at first sex** | | |
| Less than 15 years | 9,118 | 20.34% |
| Greater than 15 years | 35,719 | 79.66% |

majority (63.82%) of youth girls had media exposure and (60.56%) of youth girls were married [**Table 1**].

As shown in [**Table 2**], most of the respondents were from Malawi (14.37%) and the smallest numbers of respondent were from Comoros (3.60%). The majorities (78.87%) of the respondents were rural dwellers and nearly 51% have community-level media exposure.

## Pooled prevalence of pregnancy termination in East African countries

As shown in [**Fig 1**], the pooled prevalence of pregnancy termination in East African countries was 7.79% (95% CI: 7.54, 8.04) with the highest proportion in Uganda 12.51% (95% CI: 11.56, 13.41) and the lowest proportion was observed in Ethiopia 5.26% (95% CI: 4.39, 6.16).

## Multilevel logistic regression analysis

**Factors associated with pregnancy termination in East Africa.** *Random effect analysis.* As shown in [**Table 3**], in the null model, about 16.7% of the total variation in pregnancy termination was occurred at the community level and is attributable to the community-level factor. The highest (45.5%) PCV in the final model (Model IV) implies that both individual and community-level factors explained 45.5% of the variation in pregnancy termination across

**Table 2. Community level characteristics of the respondent in East African countries (n = 44,837).**

| Variables | Unweighted frequency | Weighted frequency | Percent |
|---|---|---|---|
| **Country** | | | |
| Burundi | 5214 | 5,505 | 12.28% |
| Ethiopia | 2575 | 2,445 | 5.45% |
| Kenya | 5710 | 5,439 | 12.13% |
| Comoros | 1584 | 1,613 | 3.60% |
| Madagascar | 4279 | 4,226 | 9.43% |
| Malawi | 6352 | 6,444 | 14.37% |
| Mozambique | 3816 | 3,935 | 8.78% |
| Rwanda | 1535 | 1,582 | 3.53% |
| Tanzania | 3068 | 3,134 | 6.99% |
| Uganda | 5256 | 5,186 | 11.57% |
| Zambia | 3457 | 3,404 | 7.59% |
| Zimbabwe | 1910 | 1,919 | 4.28% |
| **Community level poverty** | | | |
| Low level | 22,443 | 22,916 | 51.11% |
| High level | 22.313 | 21,921 | 48.89% |
| **Community level women literacy** | | | |
| Low level | 22,491 | 22,760 | 50.76% |
| High level | 22,265 | 22,077 | 49.24% |
| **Community level media exposure** | | | |
| Low level | 22,342 | 22,066 | 49.21% |
| High | 22,414 | 22,771 | 50.79% |
| **Residency** | | | |
| Urban | 10,544 | 9,476 | 21.13% |
| Rural | 34,212 | 35,361 | 78.87% |

areas. The model fitness was checked using deviance, Log-likelihood and the model with the highest Log-likelihood and the lowest deviance (Model IV) was the best-fitted model.

*Fixed effect analysis*. In the multivariable mixed effect binary logistic regression analysis, age, marital status, educational status, parity, media exposure, working status and living countries were significant determinants of pregnancy termination in East African Countries. From the multivariable mixed effect binary logistic regression result, the odds of pregnancy termination among 20–24 year old youth were 1.93 times [AOR = 1.93; 95% CI: 1.71, 2.16] higher compared to age group 15–19 years old. Women who had media exposed were 1.22 times [AOR = 1.22; 95% CI: 1.12, 1.34] higher odd of pregnancy termination compared to the counterpart. The odds of pregnancy termination in women who were married were 1.32 times [AOR = 1.32, 95% CI: 1.21, 1.43] higher than women who were single. Women who had working were 1.13 times [AOR = 1.13; 95% CI: 1.04, 1.23] higher odd of pregnancy termination compared to the counterpart.

Regarding to educational status, mothers who attained no education, primary education, and secondary education were 3.98 times [AOR = 3.98, 95% CI: 2.32, 6.81], 4.05 times [AOR = 4.05, 95% CI: 2.38, 6.88], and 2.96 times [AOR = 2.96, 95% CI: 1.74, 5.03] higher odds of pregnancy termination compared to mothers who attained higher education, respectively. Whereas, multiparous women were lowered by 15% [AOR = 0.85; 95%CI: 0.79, 0.93] the odds of pregnancy termination compared to primiparous women. Women who had sexual initiation greater or equal to 15 years old were lowered by 18% [AOR = 0.82; 95%CI: 0.74, 0.99] odd of pregnancy termination compared to less than 15 years.

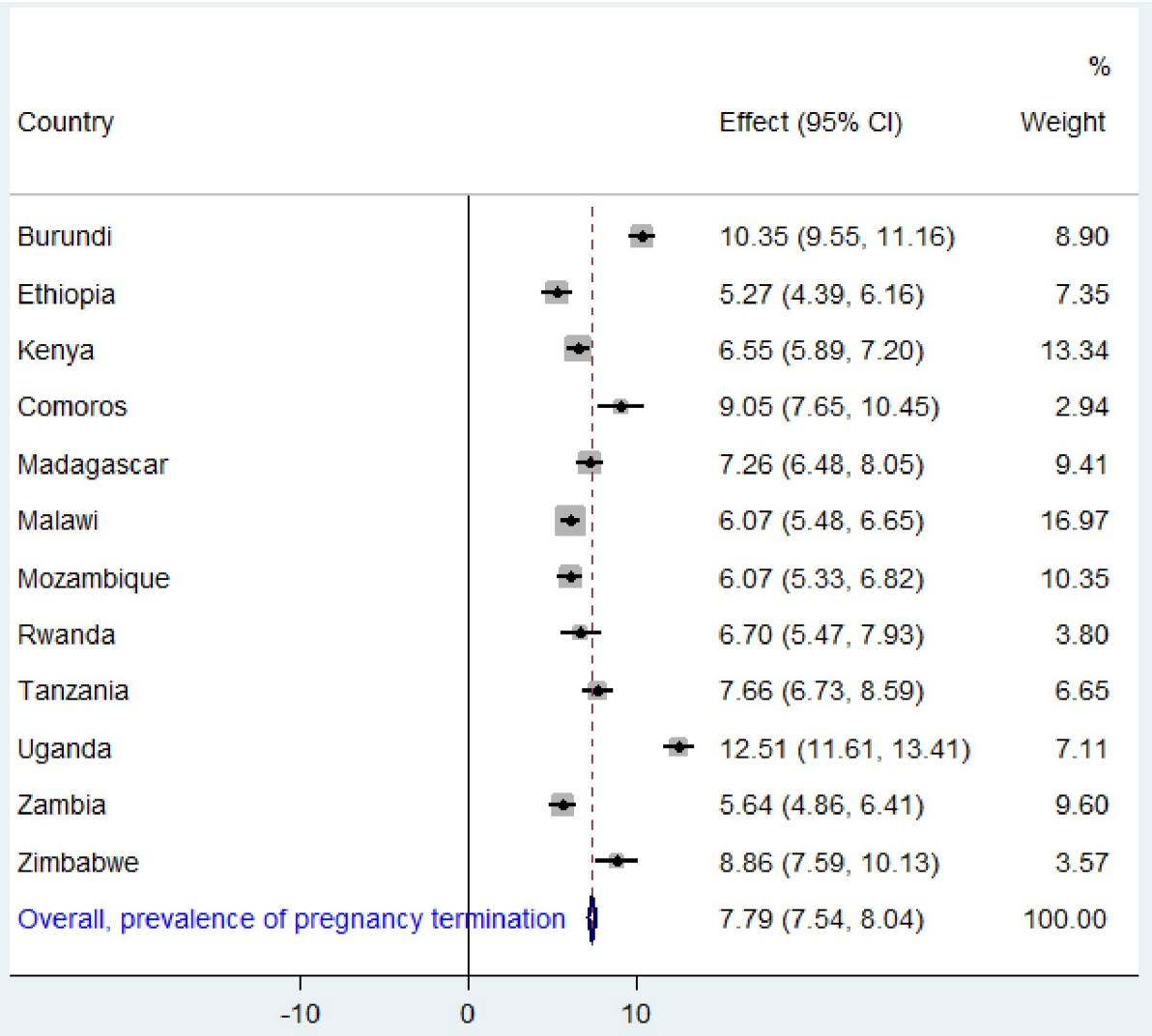

**Fig 1. Forest plot of the pregnancy termination in East African countries.**

The odds of pregnancy termination among women living in Ethiopia, Kenya, Madagascar, Malawi, Mozambique, Rwanda, and Zambia were lower by 51% [AOR = 0.49 95% CI: 0.39, 0.61], 49% [AOR = 0.51 95% CI: 0.43, 0.60], 30% [AOR = 0.70 95% CI: 0.60, 0.82], 45% [AOR = 0.55 95% CI: 0.47, 0.65], 36% [AOR = 0.64 95% CI: 0.54, 0.77], 37% [AOR = 0.63 95% CI: 0.50, 0.80], and 36% [AOR = 0.64 95% CI: 0.53, 0.77] as compared to women living in Burundi respectively. Whereas, Mothers in Uganda were 1.42 [AOR = 1.42, 95% CI: 1.24, 1.63] times higher odds of pregnancy termination compared to mothers in Burundi [Table 4].

## Discussion

The objective of this study was to examine the pooled prevalence and factors that affect pregnancy termination in 12 East African countries from 2008 to 2018 using data from recent Demographic and Health Surveys. The pooled prevalence of pregnancy termination in East African countries was 7.79% [95% CI: 7.54, 8.04] with the highest pregnancy termination in Uganda 12.51% [95% CI: 11.56, 13.41] and the lowest pregnancy termination was observed in

**Table 3. Model comparison and model fitness for multilevel logistic regression analysis.**

| Parameters | Null Model I | Model II | Model III | Model IV |
|---|---|---|---|---|
| **Random effect** | | | | |
| Community variance | 0.66[0.55,0.78] | 0.44[0.36,0.55] | 0.43[0.34,0.54] | **0.36[0.32.0.47]** |
| ICC | 16.7% | 11.9% | 11.7% | **9.8%** |
| MOR | 1.32[1.25,1.40] | 1.18[1.13,1.25] | 1.17[0.12, 1.24] | **1.13[1.11, 1.10]** |
| PCV% | 1 | 33.3% | 34.8% | **45.5%** |
| **Model comparison** | | | | |
| AIC | 23,807 | 23,583 | 23,588 | **23,315** |
| BIC | 23,825 | 23,696 | 23,736 | **23,559** |
| LLR | -11901 | -11778 | -11777 | **-11629** |
| Deviance | 23,802 | 23,556 | 23,554 | **23,258** |

NB:**AIC**: Akaike's information criterion, **BIC**: Bayesian information criterion, **LLR**: Log likelihood, **MOR**: Median odd Ratio, **ICC**: Intra-class Correlation Coefficient and **PCV** (Proportional Change in Variance).

Ethiopia 5.26% [95% CI: 4.39, 6.16].The finding was significantly greater than previous studies in Ethiopia (2.5%) [24], Nigeria (5.8%) [27], Sub-Saharan Africa (5%) [28]. This finding was lower than studies in Democratic Republic of Congo [29], Gabon (34%) [30], and Cote D'lvoire (16.1%) [30]. Due to negative socio-cultural norms, young women are frequently refused access to family planning services, which increases their risk of unwanted pregnancies that may result in abortion. The finding was significantly lower than that of study conducted in Ethiopia [19,31], Nepal [16], Africa [15]. The difference in the study population and the improvement in maternal health care service accessibility and utilization over time could be the possible explanation.

Abortion's legal situation varies considerably from country to country; a large majority of countries permit abortion in some circumstances, while others completely prohibit abortion. The majority of developed countries permit the practice. Countries set restrictions on abortion, often allowing it only in limited circumstances such as economic reasons, threats to the woman's physical or mental health, or the presence of fetal defects [32,33].

In the multilevel mixed effect logistic regression analysis age, marital status, education status, parity, age at first sex, media exposure, working status and living countries were determinants of pregnancy termination in the East Africa Countries.

This study evidenced that youth age group 20–24 years were more likely to experience pregnancy termination than youth in the age group 15–19 years. This was in line with the studies conducted in Denmark [34] and Mozambique [35]. Women who were married had a higher chance of terminating their pregnancy than women who were unmarried. It was consistent with a study finding in Africa [24,27]. That might be related to cultural social desirability, as unmarried women may be hesitant to report pregnancy termination negative stigma associated with non-marital sexual intercourse or modern contraceptive use. The odds of having pregnancy terminated were high among women who were working compared to their counterpart. This finding was supported with other studies [36,37]. This might be attributable to their priority of continuous employment and long-term professional objectives; their better consciousness of contraceptive options, particularly termination; plus their better financial ability to access illegal abortion services.

Women who had media exposure were more likely experience pregnancy termination compared to the counterpart. This finding similar with study in Mozambique [19,35]. This could be due to that media plays a vital role in broadcasting information about how and where to

**Table 4. Multivariable multilevel logistic regression analysis of both individual and community-level factors associated with pregnancy termination in East African countries.**

| Characteristics | Model I (95%CI AOR) | Model II (95%CI AOR) | Model III (95%CI AOR) | Model IV (4 95%CI AOR) |
|---|---|---|---|---|
| **Maternal age** | | | | |
| 15–19 | | 1 | | 1 |
| 20–24 | | 1.90[1.69,2.13]* | | **1.93[1.71,2.16]*** |
| **education** | | | | |
| Higher education | | 1 | | 1 |
| No education | | 3.48[2.04,5.93]* | | **3.98[2.32,6.81]*** |
| Primary education | | 3.63[2.14,6.13]* | | **4.05[2.38,6.88]*** |
| Secondary | | 2.73[1.61,4.63]* | | **2.96[1.74,5.03]*** |
| **Wealth status** | | | | |
| Rich | | 1 | | 1 |
| Poor | | 0.96 [0.87,1.05] | | 1.02[0.92,1.12] |
| Middle | | 0.98 [0.88,1.09] | | 1.01[0.91,1.13] |
| **Maternal working** | | | | |
| Not working | | 1 | | 1 |
| Working | | 1.27[1.17,1.37]* | | **1.13[1.04,1.23]*** |
| **Parity** | | | | |
| Prim parous | | 1 | | 1 |
| Multiparous | | 0.92[0.85, 1.00] | | **0.85 [0.79,0.93]*** |
| **Media exposure** | | | | |
| Not exposed | | 1 | | 1 |
| exposed | | 1.28[1.18,1.39]* | | **1.22[1.12,1.34]*** |
| **Marital status** | | | | |
| Single | | 1 | | 1 |
| Married | | 1.11[1.03,1.20]* | | **1.32[1.21, 1.43]*** |
| **Age of first sex** | | | | |
| Less than 15 year | | 1 | | **1** |
| > = 15 year | | 0.86[0.79, 0.95]* | | **0.82[0.74,0.90]*** |
| **Community level factors** | | | | |
| **Place of residency** | | | | |
| Urban | | | 1 | 1 |
| Rural | | | 0.99 [0.91, 1.09] | 0.92 [0.82,1.02] |
| **Community level poverty** | | | | |
| Low | | | 1 | 1 |
| High | | | 1.00[0.89, 1.13] | 0.99[0.87,1.12] |
| **Community level literacy** | | | | |
| Low | | | 1 | 1 |
| High | | | 1.10 [0.97, 1.24] | 1.11[0.98,1.27] |
| **Community level media exposure** | | | | |
| Low | | | 1 | 1 |
| High | | | 0.99 [0.87,1.13] | 0.95[0.85,1.08] |
| **Country** | | | | |
| Burundi | | | 1 | 1 |
| Ethiopia | | | 0.49 [0.39, 0.60]* | **0.49[0.39,0.61]*** |
| Kenya | | | 0.53 [0.45, 0.63]* | **0.51[0.43,0.60]*** |
| Comoros | | | 0.93 [0.76, 1.15] | 0.94[0.79,1.18] |
| Madagascar | | | 0.71 [0.49, 0.83]* | **0.70[0.60,0.82]*** |

*(Continued)*

**Table 4.** (Continued)

| Characteristics | Model I (95%CI AOR) | Model II (95%CI AOR) | Model III (95%CI AOR) | Model IV (4 95%CI AOR) |
|---|---|---|---|---|
| Malawi | | | 0.57 [0.49, 0.66]* | **0.55[0.47,0.65]*** |
| Mozambique | | | 0.61 [0.52, 0.73]* | **0.64[0.54,0.77]*** |
| Rwanda | | | 0.65[0.52, 0.82]* | **0.63[0.50,0.80]*** |
| Tanzania | | | 0.86 [0.73, 1.01] | 0.84[0.71,1.00] |
| Uganda | | | 1.33 [1.16, 1.51]* | **1.42[1.24,1.63]*** |
| Zambia | | | 0.57 [0.48, 0.68]* | **0.64[0.53,0.77]*** |
| Zimbabwe | | | 0.78 [0.64, 0.95]* | 0.90[0.73,1.11] |

end a pregnancy. Moreover, women who are exposed to the media may be better well-informed of abortion principles and are less likely to be stigmatized by society [38].

Regarding to educational status, women who had no education, primary education, and secondary education were higher odds of pregnancy termination compared to mothers who attained higher education. This is consistent with studies in China [39], Burkina Faso [40], and Nepal [16]. The possible explanations might be women with a higher level of education are more likely to use contraception to prevent unwanted pregnancies that are more common and can lead to pregnancy termination [28,41].

With regards to birth history, multiparous women were less likely to have pregnancy termination compared to primiparous women. This is consistent with studies in Sub-Saharan Africa [42], This may be because women without children are more likely to be teenagers, and the likelihood of unwanted pregnancies, which frequently result in abortion, has been observed to be higher among young women due to an unmet demand for family planning [35,43]. Women who started sexual activity early were more likely to terminate their pregnancy compared to their counterparts. This was consistent with studies conducted in Ghana [23]. The possible explanation for this finding is that young women who have sex at a younger age are more likely to become pregnant unexpectedly. These undesired pregnancies have been associated with a lack of or insufficient access to sexual and reproductive health services, such as contraceptives, as well as socio-cultural norms surrounding contraceptive access among young women who have their first intercourse before the age of 15 years [44,45].

## Strengths and limitations

As strength, the study relied on pooled weighted nationally representative DHS surveys from 12 East African nations, which were weighted to ensure representativeness and a valid estimate. Secondly the study used an advanced model to account for the clustering effect (multilevel logistic regression) in order to obtain a reliable standard error and estimate. Moreover, the analysis used a large sample size, which may have increased the study's power to decide the true influence of the covariates. As a limitation, the temporal relationship cannot be demonstrated due to the cross-sectional character of the data. Furthermore, because the outcome was vulnerable and depended on self-reporting, there is a chance of social desirability bias, which might lead to under-reporting.

## Conclusion

The pooled prevalence of pregnancy termination in East Africa was high in this study. Maternal age, marital status, education status, parity, age at first sex, media exposure, working status and living countries were significantly associated with pregnancy termination. The finding

provides critical information for developing health interventions to decrease unplanned pregnancies and illegal pregnancy termination. Our findings imply that additional consideration should be paid to the most vulnerable populations of youth aged 20–24 years by programmers and practitioners in order to build more inclusive, thoughtful, and responsive abortion care. It is advised that government and non-governmental organizations in East Africa countries improve sexual education and regularly enhance youth sexual and reproductive health programs targeted at youth at risk of pregnancy termination.

## Supporting information

**S1 Questionnaire.**
(DOCX)

## Acknowledgments

We would like to express our deepest thankfulness to Measure DHS, for providing the data for the study.

## Author Contributions

**Conceptualization:** Samuel Hailegebreal, Ermias Bekele Enyew, Binyam Tariku Seboka, Firehiwot Haile.

**Formal analysis:** Girma Gilano.

**Investigation:** Atsedu Endale Simegn.

**Methodology:** Samuel Hailegebreal, Ermias Bekele Enyew, Binyam Tariku Seboka, Girma Gilano, Mohammedjud Hassen Ahmed, Yosef Haile.

**Project administration:** Binyam Tariku Seboka.

**Resources:** Atsedu Endale Simegn, Girma Gilano.

**Software:** Samuel Hailegebreal, Ermias Bekele Enyew.

**Supervision:** Reta Kassa, Mohammedjud Hassen Ahmed.

**Validation:** Binyam Tariku Seboka.

**Writing – original draft:** Samuel Hailegebreal.

**Writing – review & editing:** Atsedu Endale Simegn, Reta Kassa, Mohammedjud Hassen Ahmed, Yosef Haile, Firehiwot Haile.

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
