## [Decision Letter · Decision Letter 0]

24 May 2022

PONE-D-22-07864Prevalence and associated factors of pregnancy termination among youth aged 15–24 year women in East Africa: multilevel level analysisPLOS ONE

Dear Dr. Hailegebreal,

Thank you for submitting your manuscript to PLOS ONE. After careful consideration, we feel that it has merit but does not fully meet PLOS ONE’s publication criteria as it currently stands. Therefore, we invite you to submit a revised version of the manuscript that addresses the points raised during the review process. Please submit your revised manuscript by Jul 01 2022 11:59PM. If you will need more time than this to complete your revisions, please reply to this message or contact the journal office at plosone@plos.org. Please include the following items when submitting your revised manuscript:A rebuttal letter that responds to each point raised by the academic editor and reviewer(s). You should upload this letter as a separate file labeled 'Response to Reviewers'.A marked-up copy of your manuscript that highlights changes made to the original version. You should upload this as a separate file labeled 'Revised Manuscript with Track Changes'.An unmarked version of your revised paper without tracked changes. You should upload this as a separate file labeled 'Manuscript'.

We look forward to receiving your revised manuscript.

Kind regards,

Blessing Akombi-Inyang, Ph.D.

Academic Editor

PLOS ONE

Journal Requirements:

Additional Editor Comments:

Please ensure to adequately address the reviewers' comments so as to improve the quality of your paper.

Reviewers' comments:

Reviewer's Responses to Questions

**Comments to the Author**

1. Is the manuscript technically sound, and do the data support the conclusions?

Reviewer #1: Yes

Reviewer #2: Yes

2. Has the statistical analysis been performed appropriately and rigorously? 

Reviewer #1: I Don't Know

Reviewer #2: Yes

3. Have the authors made all data underlying the findings in their manuscript fully available?

Reviewer #1: Yes

Reviewer #2: Yes

4. Is the manuscript presented in an intelligible fashion and written in standard English?

Reviewer #1: Yes

Reviewer #2: Yes

5. Review Comments to the Author

Reviewer #1: Paper by Doctor Hailegebreal S et al. treated factors associated with pregnancy termination in young women in East Africa. I would like to present comments to improve the manuscript.

[Major]

1. Table 4: Readers may be perplexed as to which model is the final model.

2. Discussion: A statement on whether abortion is legally prohibited or permitted is also desired. Readers from other African countries, especially Buddhist countries, will need this information when reading this paper.

3. Following the results of this study, do you have any recommendations on measures to prevent abortion? In particular, it would be good to have a paragraph with a message for someone specific, be it a policy maker, a young female resident or a health worker.

4. Discussion: I consider more citation need to be quoted at more sentences in the Discussion section.

5. Results: Has multicollinearity been considered in Model 4?

6. Discussion: When citing previous studies and comparing them with the results of this study, it is also important to inform the reader about the figures given in previous studies.

7. Methods: Further statements need to be made about the data source. The authors cite a website, but the description should be written in such a way that the reader can get an overview of the data sources without having to visit this website.

8. Results: Is it possible to add a diagram so that the results of the final model can be instantly understood? Even if it is not the final model, just showing the percentage of abortions in each country would be a high impact figure.

[Minor]

9. Methods: The authors write that the stratified random sampled data was weighted to stretch the numbers. Can you describe the method in more detail? So that readers who are not familiar with statistical processing can understand it to some extent.

10. Background: Background may be a little long. Could the need for this research be addressed more directly?

This study adds a certain amount of evidence to obstetrics and health administration. I consider that it would be better evidence with modifications.

Reviewer #2: Thank you for the opportunity to read the paper "Prevalence and associated factors of pregnancy termination among youth aged 15–24 year women in East Africa: multilevel level analysis". The most interesting aspect of the paper is the use of Intra-class Correlation Coefficient (ICC), Median Odds Ratio (MOR), and deviance (-2LLR) values were used to assess model comparison and fitness. Notwithstanding the paper is not clear on a number of methodological issues. I have indicated a number of them below:

In the background the authors state "Approximately 73 million induced abortions are conducted worldwide each year. One third of all unsafe abortions in Africa were performed under the most hazardous conditions, i.e. by untrained individuals utilizing harmful and invasive techniques. The authors should cite those studies they were referring to.

The authors have presented the weighted sample for each country. However, it is also important to present a table that shows the number of adolescents in each country, the number that were eligible for the study and the actual number that was used for each country.

The outcome variable of this study was "pregnancy termination", What does this statement mean? How was this variable derived? Did the authors use adolescents who were pregnant at the time of the study or those who had ever experienced adolescent pregnancy?

How was each of the individual and community level variables used in the study coded? For instance, how was access to media derived and coded? How was community women education derived and coded?

6. PLOS authors have the option to publish the peer review history of their article (what does this mean?). If published, this will include your full peer review and any attached files.

Reviewer #1: No

Reviewer #2: **Yes: **Junaid Ahmad

---

## [Author Response · Author response to Decision Letter 0]

10 Aug 2022

To: PLOS ONE 

From: Samuel Hailegebreal 

Subject: A letter Accompanying Revision in Response to Editors and Reviewer Comments

Dear Editors 

The authors would like to thank the editorial team and team of reviewers for constructive and valuable comments. The authors are very happy to submit the revised version of the manuscript entitled “Pooled prevalence and associated factors of pregnancy termination among youth aged 15–24 year women in East Africa: multilevel level analysis” for its publication in your Journal. The comments of the editors and the reviewers were highly insightful and enabled us to greatly improve the quality of our manuscript. In this revised manuscript we made substantial changes to address your concerns in a point-by-point response. We are very keen to incorporate further comments, if any, for the betterment of the final manuscript.

Point by Point Response to – Reviewer Comments 

Reviewer #1

1. Table 4: Readers may be perplexed as to which model is the final model.

Authors’ response: Thank you very much for the comment. The model fitness was evaluated using deviation and Log-likelihood, and the model with the highest Log-likelihood and lowest deviance (Model IV) was the best-fitted model, which we included in table 3.

2. Discussion: A statement on whether abortion is legally prohibited or permitted is also desired. Readers from other African countries, especially Buddhist countries, will need this information when reading this paper.

Authors’ response: Thank you, reviewer, for your valuable comment. We accepted and corrected it. (See the revised manuscript)

3. Following the results of this study, do you have any recommendations on measures to prevent abortion? In particular, it would be good to have a paragraph with a message for someone specific, be it a policy maker, a young female resident or a health worker.

Authors’ response: Thank you, reviewer, for your valuable comment. We accepted and corrected it. (See the revised manuscript in conclusion section)

4. Discussion: I consider more citation need to be quoted at more sentences in the Discussion section.

Authors’ response: Thank you, reviewer, for your valuable comment. We accepted and corrected it. (See the revised manuscript in discussion section)

5. Results: Has multicollinearity been considered in Model 4?

Authors’ response: Thank you, reviewer, for your valuable comment. We accepted and corrected it. (See the revised manuscript in data management and analysis sextion)

6. Discussion: When citing previous studies and comparing them with the results of this study, it is also important to inform the reader about the figures given in previous studies.

Authors’ response: Thank you, reviewer, for your valuable comment. We accepted and corrected it. (See the revised manuscript in discussion section)

7. Methods: Further statements need to be made about the data source. The authors cite a website, but the description should be written in such a way that the reader can get an overview of the data sources without having to visit this website.

Authors’ response: Thank you, reviewer, for your valuable comment. We accepted and corrected it. (See the revised manuscript in Method section)

8. Results: Is it possible to add a diagram so that the results of the final model can be instantly understood? Even if it is not the final model, just showing the percentage of abortions in each country would be a high impact figure.

Authors’ response: Thank you, reviewer, for your valuable comment. Model IV is the final model, as shown in table 3. We also calculated the proportion of abortions for each country and the overall prevalence in forest plot fig.1.

9. Methods: The authors write that the stratified random sampled data was weighted to stretch the numbers. Can you describe the method in more detail? So that readers who are not familiar with statistical processing can understand it to some extent.

Authors’ response: Thank you, reviewer, for your valuable comment. We accepted and corrected it. (See the revised manuscript in Method section of Sampling procedures and study population)

10. Background: Background may be a little long. Could the need for this research be addressed more directly? 

Authors’ response: Thank you, reviewer, for your valuable comment. We accepted and corrected it. (See the revised manuscript in background section)

Reviewer #2

1. Thank you for the opportunity to read the paper "Prevalence and associated factors of pregnancy termination among youth aged 15–24 year women in East Africa: multilevel level analysis". The most interesting aspect of the paper is the use of Intra-class Correlation Coefficient (ICC), Median Odds Ratio (MOR), and deviance (-2LLR) values were used to assess model comparison and fitness. Notwithstanding the paper is not clear on a number of methodological issues. I have indicated a number of them below:

Authors’ response: Thanks in advance reviewer for your critical view to advance the quality of our manuscript. We improved accordingly

2. In the background the authors state "Approximately 73 million induced abortions are conducted worldwide each year. One third of all unsafe abortions in Africa were performed under the most hazardous conditions, i.e. by untrained individuals utilizing harmful and invasive techniques. The authors should cite those studies they were referring to.

Authors’ response: Thank you, reviewer, for your valuable comment. We accepted and update accordingly

3. The authors have presented the weighted sample for each country. However, it is also important to present a table that shows the number of adolescents in each country, the number that were eligible for the study and the actual number that was used for each country.

Authors’ response: Thank you, reviewer, for your valuable comment. We accepted and corrected it. (See the revised manuscript of table.2)

4. The outcome variable of this study was "pregnancy termination", what does this statement mean? How this was variable derived? Did the authors use adolescents who were pregnant at the time of the study or those who had ever experienced adolescent pregnancy?

Authors’ response: Thank you, reviewer, for your valuable comment. Actually our study participants were not adolescent our study population were youth age 15-24 years age in east African region. The outcome variable in this study was pregnancy termination. Pregnancy termination in the DHS includes induced abortions, stillbirths and miscarriages. To derive this variable, survey participants were asked “have you ever had a pregnancy terminated?” Two responses emanated from this question “No” and “Yes”. These two responses were used to define the outcome variable in line with previous studies. The source population was all pregnant young women 15-24 years old across the east African countries whereas, all pregnant young women 15-24 years old in the selected enumeration areas were the study population. 

5. How was each of the individual and community level variables used in the study coded? For instance, how was access to media derived and coded? How was community women education derived and coded?

Authors’ response: The Individual-level variables were coded age (15-19 years and 20-24 years), educational status(No education, Primary education, Secondary education and Higher education), working status(working and not working), parity(Prim parous, and Multiparous), marital status(single, and married), wealth index(poor, middle, and rich, using principal components analysis (PCA) ), age at first sex(Less than 15 year, and >= 15 year ), and media exposure(yes/no). 

The community-level factors were community-wealth index, community-level education, community-level mass media (radio, TV and newspaper) exposure). The community variables were created by aggregation of individual level variable wealth status, education, and media exposure (see the revised manuscript in operational definition).

---

## [Editor Report · Decision Letter 1]

14 Sep 2022

Pooled prevalence and associated factors of pregnancy termination among youth aged 15–24 year women in East Africa: multilevel level analysis

PONE-D-22-07864R1

Dear Dr. Hailegebreal,

We’re pleased to inform you that your manuscript has been judged scientifically suitable for publication and will be formally accepted for publication once it meets all outstanding technical requirements.

Kind regards,

Blessing Akombi-Inyang, Ph.D.

Academic Editor

PLOS ONE

Additional Editor Comments (optional):

Please revise as follows:

- The data was received from the measure DHS program (REF) after SUBMITTING prepared concept notes about the project.

N/B: Remove the link and replace it with a reference.

- The pooled prevalence of pregnancy termination in East African countries was 7.79%, with the highest pregnancy termination in Uganda (12.51%) and the lowest in Ethiopia (5.26%).

N/B: Avoid repeating confidence interval in the discussion section when it had already been provided in the result section.

- Our findings imply that additional consideration should be GIVEN to the most vulnerable SUB-populations of youth aged 20-24 years by POLICY MAKERS AND PUBLIC HEALTH RESEARCHERS to build more inclusive, thoughtful, and responsive abortion care.

N/B: The words "programmers" and "practitioners" could mean anything. Be more specific.
---

## [Editor Report · Acceptance letter]

14 Dec 2022

PONE-D-22-07864R1 

Pooled prevalence and associated factors of pregnancy termination among youth aged 15–24 year women in East Africa: multilevel level analysis 

Dear Dr. Gele:

I'm pleased to inform you that your manuscript has been deemed suitable for publication in PLOS ONE. Congratulations! Your manuscript is now with our production department. 

Kind regards, 

on behalf of

Dr. Blessing Akombi-Inyang 

Academic Editor

PLOS ONE